# Work-Related Smartphone Use at Night and Job Satisfaction: Testing a Moderated Mediation Model of Emotional Exhaustion and Organizational Dehumanization

**DOI:** 10.3390/ijerph191710674

**Published:** 2022-08-27

**Authors:** Francis Cheung

**Affiliations:** Department of Applied Psychology, Lingnan University, Hong Kong; francischeung@ln.edu.hk

**Keywords:** smartphone use, job satisfaction, emotional exhaustion, organizational dehumanization

## Abstract

Work-related smartphone use at night has attracted substantial research attention. Surprisingly, its impact on employees’ job satisfaction is mixed. Based on the stressor–strain–outcome model, this study aims to examine whether emotional exhaustion mediates the relationship between work-related smartphone use at night and job satisfaction. Furthermore, the role of organizational dehumanization in moderating the relation between work-related smartphone use and emotional exhaustion, and the association between emotional exhaustion and job satisfaction, was examined. A total of 372 participants reported on two online surveys. Bivariate correlation results showed that work-related smartphone use was positively related to emotional exhaustion but there was no significant association between work-related smartphone use and job satisfaction. Moderated mediation analysis results suggested that organizational dehumanization (T1) did not interact with work-related smartphone use at night (T1) in predicting emotional exhaustion (T1). However, organizational dehumanization (T1) interacted with emotional exhaustion (T1) in predicting job satisfaction (T2), in which individuals who perceived higher organizational dehumanization reported lower job satisfaction under higher emotional exhaustion. The limitations and implications of this study are also discussed in this paper.

## 1. Introduction

Smartphones have become an indispensable gadget in people’s lives. Based on GSMA intelligence, as many as 5.34 billion of people or 66.9% of the total population are mobile phone users. With its embedded functions and enhanced connectivity, mobile phones have undoubtedly provided ease in everyday life. It has also reshaped the work pattern as its use in the workplace has improved employees’ work efficiency [1,2].

Although work-related smartphone use has positive impacts for employees and organizations, a growing body of research has reported negative outcomes of smartphone use, especially when used at night or after office hours [3,4,5,6,7,8]. Interestingly, prior research has often explored the relation between work-related smartphone use at night on job satisfaction, which is broadly defined as the feeling of fulfillment or contentment employees feel with their job, the findings are less conclusive as mixed results were reported. 

In order to delineate the relation between work-related smartphone use at night and job satisfaction, more research is warranted. Thus, the present study aims to apply the stressor–strain–outcome model [9] to explore the mechanism of how work-related smartphone use at night relates to job satisfaction. Furthermore, the association between work-related smartphone use at night and job outcomes often depends on the organizational context, such as work culture and organizational norms [3,4,10,11,12].

However, there is a paucity of studies addressing how other organizational characteristics that may affect the relation between work-related smartphone use. Thus, the second goal of this research is to advance the existing literature by exploring an aversive organizational feature, namely organizational dehumanization, in moderating the hypothesized work-related smartphone and job satisfaction association.

To the best of the author’s knowledge, this is the first empirical study to apply the stressor–strain–outcome model in testing the association between work-related smartphone use at night and job satisfaction. This is also the first attempt to explore the role of organizational dehumanization in the work-related smartphone use literature. The results should provide a theoretical contribution to the growing literature of the work-related smartphone use by providing a new or alternative theoretical perspective to study how the smartphone use associate with occupational well-being, and how organizational characteristics, such as organizational dehumanization, enrich the understanding of work-related smartphone use and its impacts.

### 1.1. Work-Related Smartphone Use at Night and Job Satisfaction

As discussed earlier, the first goal of the present study is to investigate the relation between work-related smartphone use at night and job satisfaction. At present, research could not conclude the impact of the use of work-related smartphone for work at night on job satisfaction [13]. Some studies report that smartphone use is positively related to employees’ job satisfaction [14], whereas other studies have revealed a negative relation [15] or no relation between the two variables [16].

The inconsistency may be due to the very nature of the smartphone use as its availability can be a mixed blessing: On the positive side, it can provide employees’ access to work by instant access to work-related information and being connected to major stakeholders, such as coworkers, supervisors, and clients [1,2]. For example, work-related smartphone could be conceptualized as a form of work resource that allow employees to complete the job more effectively, it also supports their work via increased flexibility and sense of job control [3]. Higher job autonomy and control over how the work could be performed should result in higher sense of job satisfaction.

However, since the smartphone has removed the temporal and spatial barriers from employees’ access to work-related information and work tasks, employees can continue with their work, either willingly or under the pressure by others, especially when the organization expects employees to be constantly available after work hours [2]. Previous studies often show that the use of smartphone for work purpose at night, such as responding to work emails, replying work-related messages via instant messenger apps (e.g., Whatsapp and Signal) were taxing to employees’ occupational and psychological well-being because the use of smartphone to perform their work duties at night represents a continuous drain of employees’ important resources and they could not detach from their work [3].

In a recent narrative review, Schlachter and colleagues [17] reported that the use of smartphone phone or work-related technology use during non-work time was frequently found to associate negatively with poorer occupational well-being, including lower job satisfaction. Based on the boundary theory, smartphone use extending work into private life would blur the boundaries between these life and work domains, which may impair employees’ sense of job satisfaction.

Based on the above discussion, even though previous studies cannot provide a clear indication of the relations between work-related smartphone use at night and job satisfaction, it is hypothesized that such smartphone use would be negatively related to job satisfaction. Several reasons support this hypothesis. First, high connectivity relates to work-related smartphone use at night create difficulties for employees to psychologically detach from their work [13], and a lower psychological detachment from work will likely lead to poorer occupational wellbeing, including job satisfaction [18].

Furthermore, work-related smartphone use at night potentially leads to higher role overload, in which employees must handle work- and non-work-related responsibilities simultaneously (i.e., work duty overlap with interpersonal/social roles). Role overload can occur when work and non-work domains compete for employees’ scarce resources (e.g., time and physical energy), which may lead to negative outcomes, including lower job satisfaction [15].

Finally, the use of smartphones for work purpose at night would also compete for the self-regulatory resources. Gombert, Rivkin, and Schmidt reported that by performing work duties via the smartphone, such as staying online, checking for new work messages, these work-related activities would drain the employees’ limited self-regulatory resources. These work hassles would negatively affect ones’ evaluation of their job satisfaction. Based on these observations, this study proposes the following:

**Hypothesis** **1.**
*Work-related smartphone use at night is negatively related to job satisfaction.*


### 1.2. Emotional Exhaustion as the Mediator between Work-Related Smartphone Use and Job Satisfaction

Apart from the direct effect, work-related smartphone use at night is further hypothesized to predict job satisfaction via the mediation of emotional exhaustion. The stressor–strain–outcome model is used as the theoretical backdrop of the mediation hypothesis. The stressor–strain–outcome model is a theoretical model that delineates how a work stressor predicts outcomes via the mediation of strain [9,19]. In this framework, a work stressor refers to an environmental stimulus or work condition perceived by employees as disruptive. Common work stressors include excessive workloads, lack of support, role ambiguity, and role conflict [9].

Other empirical studies have examined industry-specific stressors, such as emotional dissonance [20] and customer-related stressors [21], in the service industry. The experience of work stressors will lead to higher strain, and it creates a disruptive effect on employees’ concentration, physiology, and emotion. Thus, the strain experienced would predispose employees to negative job and occupational outcomes [22], such as lower job satisfaction, organizational citizenship behaviors, and higher intentions to leave the organization.

Based on the stressor–strain–outcome model, work-related smartphone use at night is positioned as a work stressor that will predict job strain. First, work-related smartphone use at night relates to the expectations of employees being available outside regular work hours, and they perceive it as important to “stay connected” by smartphone [23]. This results in employees’ urge to use smartphone for work purposes after typical work hours, which will drain their resources [24].

Second, work-related smartphone use at night also influences the engagement of recovery activity. Employees who have higher work-related smartphone use at night would report higher work–home interference, and they tend to engage less in recovery activities when compared with employees with lower smartphone use [6]. When employees perform their work duties by their smartphone at night, this behavior will continuously drain of their important resources. By using their smartphone to perform their work duties, they will also miss the opportunity to detach from their work and perform recovery exercise. Thus, they are likely to experience higher level of work strain, such as emotional exhaustion [25].

Emotional exhaustion, in turn, is hypothesized to negatively relate to job satisfaction. As a key dimension of the burnout construct, emotional exhaustion is characterized by the depletion of energy and is the result of enduring physical, affective, or cognitive energy drain. It reduces the ability of employees to cope with and meet their demands at work. Thus, when feeling emotionally exhausted, employees may feel less energetic and less interested in their jobs. Exhausted employees would develop a more negative view on their work environment and lower their sense of job accomplishment. They also tend to adopt avoidance or withdrawal as a coping mechanism [26].

Recent research by Breevaart and Tims [27] also suggested that, when employees are emotionally exhausted, they may be more deliberate about how they invest their resources by social crafting. In particular, instead of continuous investment of their resources on their work tasks, exhausted employees may choose to spend more time to recover and replenish their resources, such as gaining social support from their co-workers and supervisors. When exhausted employees invest their effort in replenish their resources instead of devoting their resources on the job task, their sense of job accomplishment would be negatively affected, which will lead to lower job satisfaction. The association between job satisfaction and emotional exhaustion has been confirmed in a recent meta-analysis [25].

Combining the above discussion together, work-related smartphone use will first predict emotional exhaustion, and the latter will predict job satisfaction. In other words, emotional exhaustion functions as the mediator between work-related smartphone use and job satisfaction. Thus, the following is proposed:

**Hypothesis** **2.**
*Emotional exhaustion mediates the relation between work-related smartphone use and job satisfaction.*


### 1.3. Organizational Dehumanization as a Moderator

Prior studies suggested that the effect of smartphone use may interact with external factors, such as organizational norms or cultures [2,3]. In a recent study [28], researchers reported that smartphone use after work is positively related to job burnout; however, such a relation is determined by the level of social support and organizational politics. In particular, higher organizational politics may strengthen the relations between smartphone use and job burnout.

Similarly, studies by Derks and colleagues [6] also suggested that the use of work-related smartphones is determined by organizational characteristics, such as segmentation norms (employees’ perception of their availability for work outside of work hours). For example, when employees perceive the organization promote a clear work-home boundary (segmentation), they have a lower compulsion to respond to work-related tasks related to their smartphone at night.

However, when they perceive the organization value an integration norm (permeable work-home boundary and continuous availability for work), then employees might have a stronger need to respond to work messages or tasks by using their smartphone at night. Therefore, the inclusion of organizational characteristics as boundary factor would provide a more comprehensive picture of how the organizational context influence the relation between smartphone use and outcome. In this study, the second goal is to critically examine whether organizational dehumanization moderates the above relations.

Organizational dehumanization refers to employees’ perception of being used as an instrument by their organization [29]. It describes people’s perception of other human beings as something lesser than or profoundly different from themselves [30]. When employees perceive organizational dehumanization, they believe that the organization only regard them as a tool to finish specific tasks and ensure organizational effectiveness.

When working in a dehumanizing work environment, employees usually report poorer occupational wellbeing, including lower job satisfaction [31], higher stress at work [32], and emotional exhaustion [33]. Although empirical study on the relation between organizational dehumanization on occupational wellbeing is growing, studies that investigate whether smartphone use is related to organizational dehumanization are lacking.

This study aims to examine whether organizational dehumanization moderates the effect on work-related smartphone use at night and outcome association. When facing a work stressor, organizational characteristic plays a highly important role in shaping the experience. For example, based on the conservation of resource model, employees with more organizational support tend to cope better than those who have limited access to the resources [34].

On the contrary, unsupportive, or harsh organizations, such as those that are characterized by often dehumanizing their employees, may further amplify the negative effect of work stressor because they may perceive the organization will not provide necessary support to them in performing their work tasks. Furthermore, a cold and insensitive work environment, such as when employees perceive that there is a lack of social support from coworkers and supervisors, may further exacerbate the aversive consequence of smartphone use because employees do not perceive the organization to be willing to provide any assistance to mitigate high work stress [34].

Thus, when employees perceive that they are working in a dehumanized workplace, they will have a strong urge to perform more work because the organization makes them feel important solely depending on their performance at work. Drawing from the above discussion, employees may have a stronger compulsion to use their smartphone or place a higher importance of performing their work tasks with their smartphone at night because they have a sense that their continuous performance in their work is essential to be recognized by the organization, which will result in more resource drain and, thus, higher emotional exhaustion. Thus, this study proposes the following hypothesis:

**Hypothesis** **3.**
*Organizational dehumanization moderates the relation between work-related smartphone use at night and emotional exhaustion. Particularly, employees who experience higher organizational dehumanization will report higher emotional exhaustion when using smartphones at night compared with employees with lower organizational dehumanization.*


Organizational dehumanization is also hypothesized to moderate the relation between emotional exhaustion and job satisfaction. As suggested earlier, emotionally exhausted employees usually adopt a more passive coping method and create negative work views, which usually lead to lower job satisfaction. Based on the conservation of resource model, exhausted employees should reduce their input in their job in order to conserve their depleted resources, such as limiting their energy and time in their work. However, in a dehumanizing work environment, employees are merely being treated as machinery and are readily replaceable, either by other personnel or advanced technology [30,31,32,33,34].

The value of employees to the organization is largely determined by their contribution to the overall organizational success. Thus, when employees perceive they are working in a dehumanizing work environment, in order to secure their job, they would have stronger compulsion to perform their work duties and contribute to the organization because they understand their value to the organization is largely determined by their performance at work. The frustration of employees’ basic needs, such as sense of autonomy or control will further damage their sense of job satisfaction. On this basis, this study proposes the following:

**Hypothesis** **4.**
*Organizational dehumanization moderates the relation between emotional exhaustion and job satisfaction. Specifically, the association between emotional exhaustion and job satisfaction will be stronger when the employees perceive higher organizational dehumanization when compared with employees with lower organizational dehumanization.*


Figure 1 presents the conceptual framework on the present study. 

## 2. Methods

### 2.1. Participants and Procedure

The participants were recruited using a crowdsourcing marketplace in the U.S. (Amazon Mechanical Turk: https://www.mturk.com/ (accessed on 11 January 2022)). To be eligible for the study, the participants had to be 18 years or older, working full time, and residing in the U.S. The recruitment service provider sent out email invitations with a hyperlink to the online survey. Screening for eligibility was conducted through MTurk. No personal identifiable information was asked. Before responding to the survey, participants were asked to read the research objective and to provide their consent. The web survey took about 15–20 min to complete.

Upon the completion of the survey, participants were given a code number that they used to claim their participation fees from the Internet marketplace. This study was approved by the Institutional Ethics Review Board of the affiliated university of the author. In terms of data quality, the participants recruited via online platforms paid more attention to survey instructions compared with those from traditional participant pools [35]. To further enhance data quality, two filter items were inserted in each survey. A total of 479 participants completed the survey. Among them, 36 participants who responded to the filter items incorrectly were excluded from the main analyses.

One month after the first wave of data collection, participants who had completed the first wave of data collection were approached again by MTurk for the online survey. A total of 372 participants had completed the online surveys. Among these participants, 211 (56.7%) were males, 157 (42.2%) were females, and four participants did not indicate their gender or selected non-binary/third gender. The mean age of the participants was 43.26 (SD = 10.58; range = 23–78). A total of 66.4% participants had obtained a bachelor’s degree or above, and approximately 71% of these participants were working as managers/administrators, professionals, and associate professionals.

### 2.2. Measures

**Work-related smartphone use at night (Time 1).** Work-related smartphone use was measured by the six-item scale by Cheung et al., [3]. Sample item includes, “I use my mobile phone for work purpose in the evening or after work hours.” The participants were asked to respond to a five-point Likert scale, and a higher score indicates more work-related smartphone use at night. The Cronbach’s alpha of the scale in this sample was 0.92.

**Emotional exhaustion (Time 1).** Emotional exhaustion was measured with the subscale in the Maslach Burnout Inventory [36]. The measure comprises five items. Sample item includes, “I feel emotionally drained from my work.” The participants rated whether they experienced exhausted emotion on a seven-point scale. A higher score indicates a higher level of emotional exhaustion felt by the participant. The Cronbach’s alpha of the scale was 0.94.

**Organizational dehumanization (Time 1).** Organizational dehumanization was measured by an 11-item organizational dehumanization scale by Caesens et al., [33]. Sample item includes, “If my job could be done by a machine or a robot, my organization would not hesitate to replace me by this technology.” The participants rated whether they experienced organizational dehumanization on a five-point scale. A higher score indicates a higher level of organizational dehumanization felt by the participants. The Cronbach’s alpha of the scale was 0.92.

**Job satisfaction (Time 2).** A three-item job satisfaction scale by Price and Mueller [37] was used in this study. A sample item is, “I find real enjoyment in my job.” The participants were invited to respond using a seven-point scale ranging from 1 (“strongly disagree”) to 7 (“strongly agree”). The Cronbach’s alpha of the scale was 0.91 in this sample.

**Demographics.** The participants’ gender, age, occupations, and annual income were requested.

### 2.3. Plans of Analysis

Various statistical methods will be used in the analysis. Before the major analyses, common method variance by the Harman’s one factor test and confirmatory factor analysis (CFA) was conducted to determine the impact of common method variance and to examine whether the variables are empirically distinctive. After the preliminary tests, correlation analyses was performed to describe the associations among key variables.

Structural equation modeling (SEM) was performed to evaluate the role of work-related smartphone use on job satisfaction (i.e., direct effect or through the mediation of emotional exhaustion). Finally, moderated mediation analysis was conducted to investigate whether organizational dehumanization moderates the relations between 1. work-related smartphone use and emotional exhaustion and 2. Emotional exhaustion and job satisfaction.

### 2.4. Common Method Variance

Although the current study involves two measurement time points, the self-reported nature might inflate the observed associations due to common method variance Podsakoff et al. [38]. To assess the effect of common method variance, a Harman’s one-factor test was carried out to evaluate the magnitude. Exploratory factor analysis was performed on all items, and the number of extracted factors was specified to be 1. If method variance was largely responsible for the covariance among the measures, then the extracted factor should account for 50% or more of the observed variance. The results suggest that four un-rotated factors were extracted, and the largest extracted factor accounted for 40.97% of the total variance. Therefore, the result could not be solely attributed to the CMV.

## 3. Results

### 3.1. Descriptive Statistics and Correlation

A confirmatory factor analysis (CFA) was performed to examine whether the targeted variables from the same participants are empirically distinctive. EQS 6.4 was used to perform the statistical analysis and raw data were used as the input. Items were used as the observed indicators, and they were specified to load on the corresponding latent factors. The results suggested that the four-factor model (work-related smartphone used, emotional exhaustion, organizational dehumanization, job satisfaction) fitted well with the data. Although the chi-square was significant (χ^2^ = 756.37, df = 224, *p* < 0.01), other fit indices were in acceptable range (CFI = 0.93. RMSEA = 0.08, 90% interval = 0.07 & 0.09). In other words, the CFA results suggested that the research variables were empirically distinctive.

The correlation results showed that smartphone use at night was positively related to emotional exhaustion at T1 (*r* = 0.10, *p* < 0.05); however, it was not correlated with job satisfaction at T2 (*r* = −0.01, *p* > 0.05) nor organizational dehumanization at T1 (*r* = 0.08, *p* > 0.05). Emotional exhaustion (T1) was significantly related to job satisfaction at T2 (*r* = −0.68, *p* < 0.01). Finally, organizational dehumanization was positively related to emotional exhaustion at T1 (*r* = 0.56, *p* < 0.01) but was negatively related to job satisfaction at T2 (*r* = −0.68, *p* < 0.01). Table 1 presents the details of the descriptive statistics and correlations among major variables.

### 3.2. Structural Equation Modeling

Based on the stressor–strain–outcome model, the effect of work-related smartphone use on job satisfaction would be fully mediated by emotional exhaustion (H2). However, as have been discussed earlier, work-related smartphone use should also directly predict job satisfaction, which would imply a partial mediation model (H1). Structural equation modeling analyses were performed to test these alternative models.

In Model 1, a full-mediation was proposed in which work-related smartphone use at night would first predict emotional exhaustion, and the latter would predict job satisfaction. In Model 2, a partial mediation model was proposed: A direct path from work-related smartphone use at night to job satisfaction was added on top of the full-mediation model. Across these models, organizational dehumanization was controlled. Since Model 1 and Model 2 were nested models, the Chi-square difference test were used to evaluate the model change. EQS 6.4 was used to perform the statistical analysis, and raw data were used as the input.

The results showed that, compared with the full-mediation model, the partial mediation model had lower chi-square (Model 1 = 144.36, df = 3, *p* < 0.01; Model 2 = 139.20, df = 2, *p* < 0.01). Chi-square difference test showed that the difference between Model 1 and Model 2 was significant (Δχ^2^ = 5.16, *p* < 0.05). Thus, the addition of the direct path from work-related smartphone use at night to job satisfaction had significantly reduced the overall model chi-square, thus, improving the model fit.

This finding also converged with the AIC index where the partial mediation model had lower AIC coefficient (Model 1 = 4502.93; Model 2 = 4499.77). In terms of the individual parameter estimate, the direct path from work-related smartphone use at night on job satisfaction was significant (β = 0.09, *p* < 0.05). These findings suggested that a revised stressor–strain–outcome model by incorporating a direct path from the stressor (i.e., work-related smartphone use at night) to the outcome (i.e., job satisfaction) fitted better than the full mediation model, where the effect of smartphone use on job satisfaction is only through the emotional exhaustion.

### 3.3. Testing the Moderated Mediation Model

The mediation and moderated mediation model was tested with the SPSS Preacher’s Marco (model 58). A total of 5000 bootstrapped samples were used in the estimation. In terms of the mediation model, where emotional exhaustion (T1) was hypothesized to mediate the relation between work-related smartphone use (T1) and job satisfaction (T2), result suggested the indirect effect was significant (β = −0.08, BootSE = 0.03, LLCI = −0.14, ULCI = −0.02).

Moderated mediation results suggested that smartphone use at night (T1) did not predict emotional exhaustion (T1) (β = −0.08, *p =* 0.57, LLCI = −0.39, ULCI = 0.22). However, organizational dehumanization (T1) was a significant predictor of emotional exhaustion (T1) (β = 0.66, *p <* 0.01, LLCI = 0.42, ULCI = 0.91). The interaction between smartphone use at night (T1) and organizational dehumanization (T1) on emotional exhaustion (T1) was also insignificant (β = 0.06, *p* = 0.23, LLCI = −0.04, ULCI = 0.16).

In predicting job satisfaction (T2), the results suggested that work-related smartphone use at night (T1) (β = 0.07, *p =* 0.03, LLCI = 0.01, ULCI = 0.14) and emotional exhaustion (T1) (β = −0.22, *p* = 0.01, LLCI = −0.40, ULCI = −0.05) were both significant. Although no direct relation was found between organizational dehumanization (T1) and job satisfaction (T2) (β = −0.05, *p* = 0.72, LLCI = −0.32, ULCI = 0.22), organizational dehumanization (T1) interacted with emotional exhaustion (T1) in predicting job satisfaction (T2) (β = −0.07, *p <* 0.01, LLCI = −0.12, ULCI = −0.01).

A simple slope analysis was conducted, and Figure 2 graphically presents the moderation effect. Particularly, employees with lower organizational dehumanization (−1 SD) reported consistently higher job satisfaction compared with their counterparts who have higher organizational dehumanization (+1 SD). The difference between high and low organizational dehumanization on job satisfaction is more obvious under a high emotional exhaustion condition.

Although the direct effect was significant (*β* = 0.07, SE = 0.03, *p* < 0.03, LLCI = 0.00, ULCI = 0.14), the hypothesized moderated mediation effect was insignificant, as 0 was included in the 95% confidence intervals when organizational dehumanization was −1 SD (β = −0.01, BootSE = 0.02, LLCI = −0.05, ULCI = 0.04), at mean level (β = −0.04, BootSE = 0.02, LLCI = −0.08, ULCI = 0.01), and +1 SD (β = −0.07, BootSE = 0.04, LLCI = −0.15, ULCI = 0.00). Therefore, no evidence supported the overall moderated mediation effect.

Finally, an ad hoc analysis was performed to evaluate if the organizational dehumanization moderated the association between work-related smartphone use and job satisfaction. Moderated regression analysis showed that in predicting job satisfaction, only organizational dehumanization (β = −0.42, t = −4.68, *p* < 0.01) was significant, while the main effect of work-related smartphone use at night (β = 0.18, t = 1.35, *p* > 0.05) and the interaction term (β = −0.18, t = 1.10, *p* > 0.05) were not significant. Based on the moderated regression result, there was no statistical support of organizational dehumanization functioned as a moderator between work-related smartphone use at night and job satisfaction.

## 4. Discussion

On the basis of the stressor–strain–outcome model, this study attempted to delineate the association between work-related smartphone use at night and job satisfaction via the mediation effect of emotional exhaustion. Furthermore, the role of organizational dehumanization as a moderator was examined.

Contrary to the hypothesis, bivariate correlation showed that work-related smartphone use at night was not significantly related to job satisfaction. Interestingly, in the subsequent structural equation modeling and moderated mediation analysis, when the use of work-related smartphone at night was used to predict job satisfaction in conjunction with other variables, including emotional exhaustion and organizational dehumanization, work-related smartphone at night was indeed found to be a significant predictor, albeit the effect was small, especially when compared to a more salient predictor, such as emotional exhaustion.

This finding provides additional evidence to the existing literature that smartphone use after work is a double-edge sword and the effect is less straightforward. On the one hand, smartphone use allows employees to continue to perform their work roles or duties, which may enhance their sense of control or overall sense of accomplishment, resulting in higher job satisfaction. On the other hand, the continuous involvement at work requires different resources input (e.g., cognitive resources), in which when the employees do not have enough replenishment, the loss of resources will become a strain (i.e., emotional exhaustion in this study).

This argument is in line with the recent finding that indicated that, when employees constantly handle work-related issues with their smartphone after work, it consumes a great deal of employees’ self-control resources for task fulfilment, and employees have difficulty in mentally disengaging from work at night [4,6]. Thus, the benefit of work-related smartphone use at night may be offset by the potential drawback of the smartphone use.

As hypothesized, bivariate correlation analysis showed that work-related smartphone use at night was positively related to emotional exhaustion. The finding was in line with other earlier studies where higher smartphone use at night contributes to higher emotional exhaustion [39,40,41]. Such an association may be related to the continuous investment of important cognitive and emotional resources for work purposes. In addition, the work involvement also hinders the employees from engaging in recovery activities, which may potentially lead to higher emotional exhaustion.

However, it is worthy to note that, when other variables (e.g., organizational dehumanization) were considered (i.e., moderated mediation analysis) in the prediction of emotional exhaustion, the role of work-related smartphone use at night became non-significant. A likely reason is the strong correlation between organizational dehumanization and emotional exhaustion (r = 0.56, *p* < 0.01) found in this study. Thus, when the effect of organizational dehumanization is considered, the unique contribution of work-related smartphone use in predicting emotional exhaustion became very limited.

In this study, a new finding relates to the relationship between smartphone use at night and organizational dehumanization. Work-related smartphone use at night is expected to be significantly related to organizational dehumanization, and the latter will also magnify the relation between smartphone use and emotional exhaustion. The results suggested that these hypotheses were not supported. Instead, the only significant moderation effect of organizational dehumanization was found in the association between emotional exhaustion and job satisfaction.

That is, emotional exhaustion was negatively related to job satisfaction, and employees who perceived higher organizational dehumanization tended to report lower job satisfaction when facing emotional exhaustion compared with those who perceived lower organizational dehumanization. Different from other organizational characteristics, such as organizational norms or perceived organizational support, the role of organizational dehumanization on smartphone use at night does not have a clear empirical support.

### 4.1. Implications

This study explored how smartphone use for work at night relates to emotional exhaustion and job satisfaction. Although a small positive effect of smartphone use was found on job satisfaction, a stronger relation between work-related smartphone use at night and emotional exhaustion was more evident. As the negative effect tends to overshadow the potential positive, albeit weak effect of job satisfaction, organizations should limit the hours used to reduce emotional exhaustion to better support the occupational health of their employees.

For example, organizations, such as Volkswagen, have configured its servers to stop routing emails to employee accounts from night to morning to minimize their employees to perform work duties with their smartphone or other devices. From employees’ point of views, they may also create a clearer boundary to disengage oneself from performing their work duties via smartphone at night. Practical ways to achieve this goal include disabling push notifications on phones or setting a reasonable screen time for work purposes at night. These behaviors allow employees to reduce the overall time to work with their smartphones at night.

The results showed that organizational dehumanization is significantly related to job dissatisfaction and emotional exhaustion. Thus, organizations should strive to reduce employees’ perception of dehumanization. For example, organizations should value the contribution of their employees toward the overall organizational effectiveness through different means, such as acknowledging employees input by providing awards and incentive.

Leaders should also pay attention to employee psychological wellbeing but not solely on their work performance. The use of periodical staff opinion survey, for example, could provide invaluable information of employees’ wellbeing and their views toward the organizations on a regular basis. Wherever possible, organizations should also establish channels that can facilitate open communication between management and employees and stop unwanted miscommunication among stakeholders.

### 4.2. Limitations and Future Studies

The current findings should be interpreted with caution due to several limitations. First, the study relied on self-reported measures in two time points, in which all data were gathered from the same source (i.e., participants’ data). Hence, common method variance might have inflated the observed associations [38]. Future studies should consider including data from other sources (e.g., ratings by co-workers) for external validation.

Second, only mechanical dehumanization was included in the study, and animalistic dehumanization was not considered. Similar to mechanical dehumanization, unfriendly treatment under animalistic dehumanization may create negative sentiments, and such negative emotions may also impair occupational wellbeing and job outcomes. Future studies may further explore how the two forms of organizational dehumanization relate to occupational health and wellbeing by examining their main effect or as moderators in smartphone use studies.

Third, job characteristics, such as task difficulties and/or overall job demands that may promote work-related smartphone use at night, were not measured. For example, when employees’ workload is high, they might choose to perform their work duties via smartphone use, even though they have left the office or is already night time. Future studies should consider the work and occupational characteristics of employees to tease out the potential confounding effects of work-related smartphone use at night.

Fourth, although the work-related smartphone use scale has clearly outlined the work tasks or the nature of the work-related smartphone use at night, whether the use is straightly imposed by the organization or the use is on a volitional or voluntary basis is not known. Such difference may better delineate the effect of work-related smartphone use at night. For example, when the employees choose to use the smartphone to perform their work tasks at night, they may find this behavior more acceptable and under control, which may lead to better occupational outcomes.

Conversely, when the employees perform their work tasks by their mobile phone at night out of organizational expectation or compulsion, they may find this as a work hassle, which may lead to more aversive well-being outcomes. Thus, future study might consider including the motivation of the work-related smartphone use at night to better delineate the purpose of the smartphone use. The differentiation of the motives may also help to better depict how the use of work-related smartphone at night relates to the occupational well-being.

## 5. Conclusions

In a nutshell, the results showed that the bivariate relation between work-related smartphone use at night and emotional exhaustion was significant; however, the effect was only modest. However, there was indeed no bivariate relation between work-related smartphone use and job satisfaction. Concerning the mediation effect of emotional exhaustion between work-related smartphone use and job satisfaction, structural equation modeling showed that, even though work-related smartphone use could significantly predict job satisfaction, the effect was small when compared to other factors, including emotional exhaustion and organizational dehumanization.

Concerning the moderation hypotheses, organizational dehumanization was found to moderate the relation between emotional exhaustion and job satisfaction). Therefore, the role of organizational dehumanization as a moderator was not strong. Although in the moderated mediation result, work-related smartphone use was a significant predictor of job satisfaction, the effect was small and was overshadowed by other predictors being considered simultaneously (i.e., organizational dehumanization).

Across earlier studies, the effect size between work-related smartphone and emotional exhaustion was typically small, e.g., [3,27,39,40,41], and the relation between work-related smartphone use and job satisfaction was inconsistent, future research is needed to unveil the “practical paradox of technology” proposed by Ter Hoeven and colleagues [8]. In particular, Ter Hoeven et al. [8] suggested that, when evaluating the impact of how communication technology, including smartphone use in influencing employees’ well-being, it is important to consider both the communication technology resources and demands, which would contribute to the employees’ well-being.

As discussed earlier, work-related smartphone use could provide accessibility and efficiency for workers, which may enhance employees’ overall well-being (e.g., job satisfaction). However, the use of smartphones for work at night also presents challenges to employees as they may encounter more unpredictability and work interruption. For example, while the fast information exchange and the expectation of a reply allow employees to continue to perform their work tasks, it may, at the same time, deplete the employees’ energy.

Therefore, the use of smartphones for work will have an impact on employee well-being through the interplay of both technology resources and demands. In this study, the use of the stressor–strain–outcome model is more inclined to depict the association on the communication demand aspect while the more facilitating aspect of technology resource is not emphasized. In order to fully understand how work-related smartphone use at night would predict well-being, future studies may adopt the approach outlined in the Ter Hoeven et al. model by considering different pathways to evaluate how the resources and demands predict occupational well-being. It is equally important to further explore how the individual and organizational characteristics moderate the effects of smartphone use on well-being.

This study contributes to the smartphone-use literature in two ways: (1) by exploring its relations with job satisfaction via the application of the stressor–strain–outcome model, and (2) to evaluate the role of boundary role of organizational dehumanization smartphone use and its association with occupational well-being. Apart from the commonly used frameworks, such as the conservation of resource model, job-demand–resource model, and boundary theory, this study provides initial evidence that a revised stressor–strain–outcome model could be applicable to understand the impact of work-related smartphone use at night on occupational well-being.

In terms of organizational dehumanization, although there was no significant evidence to suggest that this organizational characteristic moderates the relation between work-related smartphone use and the outcomes (i.e., job satisfaction and emotional exhaustion), organizational dehumanization was found to be significantly related to emotional exhaustion and also interacted with emotional exhaustion in predicting job satisfaction. Therefore, it is clear that organizational dehumanization plays a crucial role in affecting employee well-being. Given that organizational dehumanization negatively affects occupational health (i.e., emotional exhaustion) and job outcome (i.e., job satisfaction), organizations should implement effective strategies to minimize the perception of organizational dehumanization.

## Figures and Tables

**Figure 1 ijerph-19-10674-f001:**
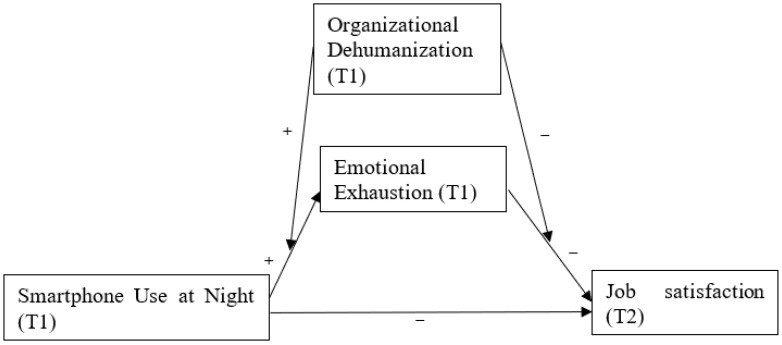
Conceptual moderated mediation model.

**Figure 2 ijerph-19-10674-f002:**
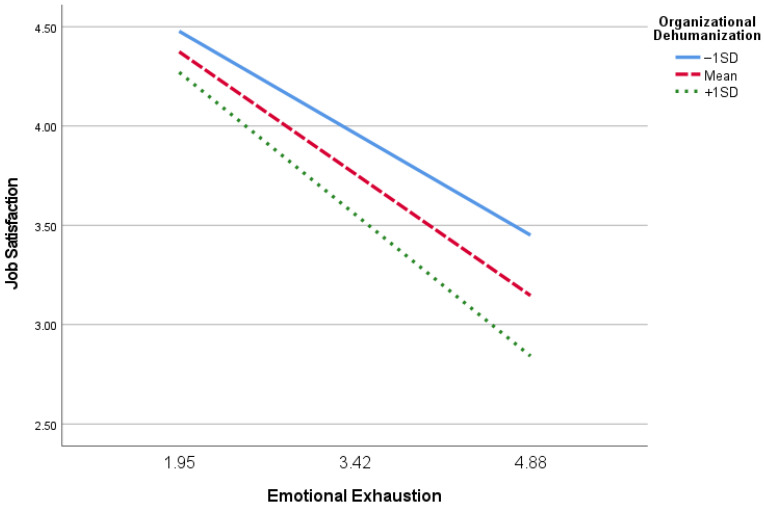
Moderation effect of organizational dehumanization on the link between emotional exhaustion and job satisfaction.

**Table 1 ijerph-19-10674-t001:** Descriptive statistics and correlation.

	M	SD	1	2	3	4	5	6
Gender	-	-	-					
2. Age	43.02	10.38	0.06	-				
3. Work-related Smartphone Use (T1)	2.32	1.21	−0.10 *	−0.06	(0.92)			
4. Emotional exhaustion (T1)	3.71	1.09	0.04	−0.08	0.10 *	(0.94)		
5. Organizational dehumanization (T1)	3.42	1.41	0.01	−0.04	0.08	0.56 **	(0.92)	
6. Job satisfaction (T2)	2.79	0.97	0.08	0.05	−0.01	−0.68 **	−0.50 **	(0.94)

Note: * *p* < 0.05, ** *p* < 0.01, α coefficient on the diagonal.

## Data Availability

The datasets generated during and or analyzed during the current study are available from the corresponding author on reasonable request.

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
