# Peer review of "Work-Related Smartphone Use at Night and Job Satisfaction: Testing a Moderated Mediation Model of Emotional Exhaustion and Organizational Dehumanization"

_ijerph, 2022, doi:10.3390/ijerph191710674_

Round 1
Reviewer 1 Report
1/ In my opinion the title formula is not correct, it is incompete. There is no term defining the analysis of the impact of the analyzed categories on each other.
2/The key problem in the presented article is the insufficient explantion of the essence of the concept of "smartphone use at night", also "worked related smartphone use at night". It is not known what the nature of the work is, whether it is voluntary work or work ordered by the employer obligatorily, why exactly at night (whether it results from the work system imposed in the organization or for another reason), how long the work is performed using a smartphone whether the use of a smartphone at night is intended only for professional tasks., or is this time also intended for other purposes. Moreover point 1.3 sounds: "Smartphone use after work....", which further complicates the understanding of the author's intentions.
3/ The main aim of the research is also not clear. In the abstract we find information as follows: "This study aims to examine whether emotional exhaustion mediates the relationship between work-related smartphone use at night and job satisfaction. Furthermore, the role of organizational dehumanization in moderating the mediation effect was examined". In the title of the article, "the organizational dehumanization" is the key moderator. It's necessary to clarify this.
4/ It is necessary to highlight the clearly accepted definitions of all key terms in order to minimize the risk of their different interpretations.
6/ Conlusions are too poor. It is worth adding the final, summary conclusions resulting from the verification of research hypotheses and recommended directions for further research.
Author Response
1/ In my opinion the title formula is not correct, it is incompete. There is no term defining the analysis of the impact of the analyzed categories on each other.
Response: Thank you for the comment. Based on your suggestion, I have revised the title into “Worked related smartphone use at night and job satisfaction: Testing the moderated mediation model of emotional exhaustion and organizational dehumanization”. I hope the change could better reflect the objectives of the study: 1. Testing the association between work-related smartphone and job satisfaction, 2. Testing the moderating and mediating role of emotional exhaustion and organizational dehumanization.
2/The key problem in the presented article is the insufficient explantion of the essence of the concept of "smartphone use at night", also "worked related smartphone use at night". It is not known what the nature of the work is, whether it is voluntary work or work ordered by the employer obligatorily, why exactly at night (whether it results from the work system imposed in the organization or for another reason), how long the work is performed using a smartphone whether the use of a smartphone at night is intended only for professional tasks., or is this time also intended for other purposes. Moreover point 1.3 sounds: "Smartphone use after work....", which further complicates the understanding of the author's intentions.
Response: Thank you for the comment. In the revised manuscript (p.2), more detailed explanation has been provided to illustrate the nature of work-related smartphone use at night (e.g. replying work-related messages via instant messengers, reply work-related emails, etc) and its potential impact on employees’ occupational well-being. Throughout the manuscript, I have also standardized the use of “work-related smartphone use at night” in order to minimize conceptual confusion for readers.
3/ The main aim of the research is also not clear. In the abstract we find information as follows: "This study aims to examine whether emotional exhaustion mediates the relationship between work-related smartphone use at night and job satisfaction. Furthermore, the role of organizational dehumanization in moderating the mediation effect was examined". In the title of the article, "the organizational dehumanization" is the key moderator. It's necessary to clarify this.
Response: Thank you for the comment. The abstract has been revised to better reflect the research objectives.
4/ It is necessary to highlight the clearly accepted definitions of all key terms in order to minimize the risk of their different interpretations.
Response: Thank you for the comment. All key constructs have been defined in the revised manuscript.
6/ Conlusions are too poor. It is worth adding the final, summary conclusions resulting from the verification of research hypotheses and recommended directions for further research.
Response: Thank you for the comment. Based on your comment, I have provided a brief summary of the verification of research hypotheses and also provide more detailed discussion on future research.
Reviewer 2 Report
Dear author,
I appreciate having the opportunity to review the manuscript entitled “Worked related smartphone use at night on emotional exhaustion and job satisfaction: Testing the moderating effect of organizational dehumanization” (ijerph-1828228).
This research investigated whether emotional exhaustion mediates the association between work-related smartphone use at night and job satisfaction. In addition, the role of organizational dehumanization in moderating the mediation effect was tested. A total of 372 respondents participated in two-wave online surveys. Although the authors have made considerable efforts to develop this paper, however, I believe that the current version of manuscript should be improved through significant revision and re-writing. I want to provide some suggestions for the improvement of this paper as follows.
[1] Introduction
- I think that the overall structure and writing of introduction part are not clear and well-aligned because it is not easy to catch what the research questions or and strategies to deal with are in this paper. Please clearly describe the exact research gaps relying on previous studies and theories. As you already knew, the introduction section is one of the most important parts to not only draw attentions of readers but also provide guidelines for them to facilitate a clear understanding of the paper.
[2] Theories and hypotheses
- Although this paper dealt with interesting phenomena, it did not provide adequate theoretical background and support for the development of its hypotheses. This is the most critical limitation of this paper. Please clearly explain the elaborate theoretical and logical support for the development of its hypotheses.
[3] Method
- I recommend the author to utilize statistical techniques that are more popular to test structural equation model (SEM) such as Mplus, Amos etc.
- The overall structure of your research is based on two-wave time lagged data. Thus, I think that this paper should provide adequate explanations about the result of confirmative factor analysis (CFA) whether the research variables from same respondents are empirically distinctive.
[4] Strengths and Limitations of the Study
- Although the authors have attempted to explain the contributions and implications of the paper, I think that the overall quality of the explanations is low. Please provide more elaborated explanations to demonstrate its theoretical and practical contributions.
I wish these comment may help you to improve your paper. Good luck.
Author Response
[1] Introduction
- I think that the overall structure and writing of introduction part are not clear and well-aligned because it is not easy to catch what the research questions or and strategies to deal with are in this paper. Please clearly describe the exact research gaps relying on previous studies and theories. As you already knew, the introduction section is one of the most important parts to not only draw attentions of readers but also provide guidelines for them to facilitate a clear understanding of the paper.
Response: Thank you very much. Based on your comment, I have revised the introduction by introducing relevant theories and to discuss more clearly on the research gap (pp. 1-2). In particular, I discussed the mixed findings of work-related smartphone use at night and job satisfaction and to propose the use of stressor-strain-outcome model as a potential way to understand this phenomenon. Besides, I have also introduced the idea of testing organizational dehumanization as the moderator.
[2] Theories and hypotheses
- Although this paper dealt with interesting phenomena, it did not provide adequate theoretical background and support for the development of its hypotheses. This is the most critical limitation of this paper. Please clearly explain the elaborate theoretical and logical support for the development of its hypotheses.
Response: Thank you for your comment. I have provided additional discussion in relevant sections so that more theories and relevant findings are used to support the hypothesis development.
[3] Method
- I recommend the author to utilize statistical techniques that are more popular to test structural equation model (SEM) such as Mplus, Amos etc.
Response: Thank you for the comment and advice. SEM is a more sophisticated and powerful tool in dealing with models with latent factors. But since all the variables in this study are directly observable and no latent factor is involved, therefore, I believe Preacher’s Macro analysis should be suitable statistical tool to perform the analysis.
- The overall structure of your research is based on two-wave time lagged data. Thus, I think that this paper should provide adequate explanations about the result of confirmative factor analysis (CFA) whether the research variables from same respondents are empirically distinctive.
Response: Thank you very much for the comment. A confirmatory factor analysis was performed to test the four-factor model (work-related smartphone use, emotional exhaustion, job satisfaction, organizational dehumanization) with the corresponding items. Results suggested that the four-factor model fitted the data reasonably well. Results were presented on p.
Reviewer 3 Report
This study aimed to examine the effect of the work-related use of smartphones at night on emotional exhaustion and job satisfaction based on the stressor-strain-outcome model. Furthermore, it intended to investigate the moderating effect of organizational dehumanization as the boundary condition of the organization on the relationships among smartphone use at night, emotional exhaustion, and job satisfaction. While there are several strengths, there are also several issues that should be addressed to make this manuscript improved.
1. Hypothesis 5b should be rewritten since it is about the moderating effect of organizational dehumanization on the relationship between emotional exhaustion and job satisfaction, not the relationship between smartphone use at night.
2. While the author(s) did not include the moderating effect of organizational dehumanization on the relationship between smartphone use at night and job satisfaction, it might provide more implications if the moderating effect of organizational dehumanization on the relationship between smartphone use at night and job satisfaction.
3. The sources of the measures are not provided. If the author(s) did not develop the measures of the variables, it is more appropriate to provide the sources of the measures of each variable.
4. The findings and discussions are not consistent.
For example on page 6, "The results suggested that smartphone use at night (T1) did not predict emotional exhaustion (T1) (β = −.08, p =.57, LLCI = −.39, ULCI = .22)." However, on page 7, "As hypothesized, smartphone use at night was positively related to emotional exhaustion. " and "The finding was in line with other earlier studies where higher smartphone use at night contributes to higher emotional exhaustion (e.g., 5,6). Such association may be related to the continuous investment of important cognitive and emotional resources for work purposes. In addition, the work involvement also hinders the employees from engaging in recovery activities, which may potentially lead to higher emotional exhaustion. "
-> If the hypothesis on the relationship between smartphone use at night and emotional exhaustion was not supported, the discussion on page 7 should be totally rewritten.
5. Non-significant correlation and significant results, and vice versa.
The correlations between smartphone use and emotional exhaustion and job satisfaction are the main frame of this study. The correlation between smartphone use and emotional exhaustion is relatively small and statistically significant (r=.10, p<.05) and the correlation between smartphone use and job satisfaction is smaller, statistically non-significant, and even negative (r=-.01, p=ns). However, the hypothesis on the relationship between smartphone use and emotional exhaustion was not statistically significant while the hypothesis on the relationship between smartphone use and job satisfaction was statistically significant. The positive and statistically significant effect of smartphone use on job satisfaction might not be the actual effect but the confounding effect. Considering the small correlations and the inconsistent results with the correlations, it is not sure whether the findings of this study are reliable.
Moreover, as the author(s) discussed, the observed relationship might have been inflated because the study relied on self-reported measures. Even with the possibility of inflation of the relationship between variables, the correlations between smartphone use and emotional exhaustion/job satisfaction are relatively negligible.
With that said, the theoretical contribution of this study is questionable.
6. On page 4, the following sentence might not be very consistent with the previous sentence.
"Thus, work-related smartphone use at night is positively related to emotional exhaustion."
Especially, "Thus" is not appropriate conjunction, considering the contents of the previous sentence.
7. On page 4, the following part is not relevant to this paper at all. Please check the manuscript carefully and revise it.
"Thus, a moderated mediation model is proposed in which job insecurity serves as a mediator and employee resilience functions as a moderator~"
Author Response
This study aimed to examine the effect of the work-related use of smartphones at night on emotional exhaustion and job satisfaction based on the stressor-strain-outcome model. Furthermore, it intended to investigate the moderating effect of organizational dehumanization as the boundary condition of the organization on the relationships among smartphone use at night, emotional exhaustion, and job satisfaction. While there are several strengths, there are also several issues that should be addressed to make this manuscript improved.
- Hypothesis 5b should be rewritten since it is about the moderating effect of organizational dehumanization on the relationship between emotional exhaustion and job satisfaction, not the relationship between smartphone use at night.
Response: Thank you for your comment. Hypothesis 5b was revised accordingly.
- While the author(s) did not include the moderating effect of organizational dehumanization on the relationship between smartphone use at night and job satisfaction, it might provide more implications if the moderating effect of organizational dehumanization on the relationship between smartphone use at night and job satisfaction.
Response: Thank you for your comment. It is an interesting idea and I have included an ad-hoc analysis to test the potential moderating effect of organizational dehumanization on work-related smartphone use at night and job satisfaction. Results did not support the moderating effect. This finding was also included in the discussion.
- The sources of the measures are not provided. If the author(s) did not develop the measures of the variables, it is more appropriate to provide the sources of the measures of each variable.
Response: Thank you for the comment. Indeed, the source / reference was included in the measure section in the numeric form. However, I have inserted the name of authors in the section to better illustrate the source for the readers’ reference.
- The findings and discussions are not consistent.
For example on page 6, "The results suggested that smartphone use at night (T1) did not predict emotional exhaustion (T1) (β = −.08, p =.57, LLCI = −.39, ULCI = .22)." However, on page 7, "As hypothesized, smartphone use at night was positively related to emotional exhaustion. " and "The finding was in line with other earlier studies where higher smartphone use at night contributes to higher emotional exhaustion (e.g., 5,6). Such association may be related to the continuous investment of important cognitive and emotional resources for work purposes. In addition, the work involvement also hinders the employees from engaging in recovery activities, which may potentially lead to higher emotional exhaustion. "
-> If the hypothesis on the relationship between smartphone use at night and emotional exhaustion was not supported, the discussion on page 7 should be totally rewritten.
Response: Thank you for the comment. The discussion is mostly derived from the testing of the bivariate correlation. In order to avoid the inconsistency of the findings and the discussion, I have specified more clearly the observation is “based on the bivariate correlation results”.
- Non-significant correlation and significant results, and vice versa.
The correlations between smartphone use and emotional exhaustion and job satisfaction are the main frame of this study. The correlation between smartphone use and emotional exhaustion is relatively small and statistically significant (r=.10, p<.05) and the correlation between smartphone use and job satisfaction is smaller, statistically non-significant, and even negative (r=-.01, p=ns). However, the hypothesis on the relationship between smartphone use and emotional exhaustion was not statistically significant while the hypothesis on the relationship between smartphone use and job satisfaction was statistically significant. The positive and statistically significant effect of smartphone use on job satisfaction might not be the actual effect but the confounding effect. Considering the small correlations and the inconsistent results with the correlations, it is not sure whether the findings of this study are reliable.
Moreover, as the author(s) discussed, the observed relationship might have been inflated because the study relied on self-reported measures. Even with the possibility of inflation of the relationship between variables, the correlations between smartphone use and emotional exhaustion/job satisfaction are relatively negligible.
With that said, the theoretical contribution of this study is questionable.
Response: Thank you for your comment. After getting this comment, I have checked and re-run all the analyses and to evaluate if there is any misrepresentation of the results. All the findings were correctly presented. The discrepancies of significant / non-significant results occurred when we compare the bivariate correlation between the key observed variables (e.g. work-related smartphone use at night, emotional exhaustion) and the subsequent testing with PROCESS Marco where multiple variables were considered simultaneously (e.g. work-related smartphone use at night + organizational dehumanization + interaction between work-related smartphone use at night & organizational dehumanization on emotional exhaustion). It is interesting to note that the actual effect size between work related smartphone use and emotional exhaustion in previous studies is mostly small to medium, and the correlation between work relate smartphone use and job satisfaction is mixed. Therefore, even though the correlation between work-related smartphone use and emotional exhaustion was only modest, it did not deviate too much from the previous findings. The theoretical contribution of the present paper is to provide a new perspective by considering the stressor-strain-outcome model to look at the impact of work-related smartphone use, and to also consider the potential effect of organizational dehumanization in the prediction. In the hindsight, the stressor-strain-outcome model might have only covered the “negative aspect” of the technical paradox outlined by Ter Hoeven et al, and I have also explained how this selection may affect the prediction of the work-related smartphone use (pp.10-11).
- On page 4, the following sentence might not be very consistent with the previous sentence.
"Thus, work-related smartphone use at night is positively related to emotional exhaustion."
Especially, "Thus" is not appropriate conjunction, considering the contents of the previous sentence.
Response: thank you for the comment. The captioned sentence was removed in order to minimize the confusion caused.
- On page 4, the following part is not relevant to this paper at all. Please check the manuscript carefully and revise it.
"Thus, a moderated mediation model is proposed in which job insecurity serves as a mediator and employee resilience functions as a moderator~"
Response: Thank you for the comment. The captioned sentence was removed in order to minimize the confusion caused.
Round 2
Reviewer 2 Report
Dear authors,
[1] Introduction
- I think that the overall structure and writing of introduction part are "still" not clear and well-aligned
- Still, it is difficult to understand the main points and contributions of this paper from the introduction part. This is the main reason why I want to reject this paper.
[2] Theories and hypotheses
- In the second round, I think that the authors did not reflect my suggestions on the quality of theory part. This paper still did not provide adequate theoretical background and support for the development of its hypotheses.
[3] Method
- I believe that this paper should provide additional chi-square different tests to compare the main model to its alternative models.
Author Response
[1] Introduction
- I think that the overall structure and writing of introduction part are "still" not clear and well-aligned
- Still, it is difficult to understand the main points and contributions of this paper from the introduction part. This is the main reason why I want to reject this paper.
Response: Thank you very much for the comment. In the revision, I have re-organized the structure of the whole introduction section. In particular, the sequence is now in line with the main research objectives: 1. To investigate the association between work-related smartphone at night and job satisfaction; 2. To investigate the organizational dehumanization role on the work-related smartphone use at night and job satisfaction. Compared to the previous submissions, materials have been streamlined and are aligned with the two overarching themes of the present study (e.g. I have removed section and hypotheses which are not aligned with the revised direction). I have also highlighted the potential contribution of the research paper at the beginning of the introduction and in the conclusion.
[2] Theories and hypotheses
- In the second round, I think that the authors did not reflect my suggestions on the quality of theory part. This paper still did not provide adequate theoretical background and support for the development of its hypotheses.
Response: Thank you for your comment. I have incorporated additional discussion in the introduction to support the hypothesis development. Relevant theory, such as boundary theory / conservation of resource theory, and other relevant research findings have been incorporated to support the hypothesis development.
[3] Method
- I believe that this paper should provide additional chi-square different tests to compare the main model to its alternative models.
Response: Thank you very much for the comments. Structural equation model analyses were included in the revised manuscript. In particular, using the chi-square difference tests, two stressor-strain-outcome models were compared (full mediation model & partial mediation model). Results suggested that the partial mediation model (work-related smartphone at night predicted job satisfaction, both directly and via the mediation of emotional exhaustion) fitted the data better when compared to the other two models.